# Attend to who you are: Supervising self-attention for Keypoint Detection and Instance-Aware Association

## Abstract

This paper presents a new method to solve keypoint detection and instance association by using Transformer. For bottom-up multi-person pose estimation models, they need to detect keypoints and learn associative information between keypoints. We argue that these problems can be entirely solved by Transformer. Specifically, the self-attention in Transformer measures dependencies between any pair of locations, which can provide association information for keypoints grouping. However, the naive attention patterns are still not subjectively controlled, so there is no guarantee that the keypoints will always attend to the instances to which they belong. To address it we propose a novel approach of supervising self-attention for multi-person keypoint detection and instance association. By using instance masks to supervise self-attention to be instance-aware, we can assign the detected keypoints to their instances based on the pairwise attention scores, without using pre-defined offset vector fields or embedding like CNN-based bottom-up models. An additional benefit of our method is that the instance segmentation results of any number of people can be directly obtained from the supervised attention matrix, thereby simplifying the pixel assignment pipeline. The experiments on the COCO multi-person keypoint detection challenge and person instance segmentation task demonstrate the effectiveness and simplicity of the proposed method, and show a promising way to control self-attention behavior for specific purposes.

## 1 Introduction

Multi-person pose estimation approaches usually can be classified into two schemes: top-down or bottom-up. Unlike the top-down scheme that converts the pipeline into two independent tasks – detection and single-pose estimation, the bottom-up scheme is confronted with more challenging problems. An unknown number of persons with any scale, posture, or occlusion condition may appear at any location of the input image. The bottom-up approaches need to detect all body joints first and group them into instances second. In the typical systems such as DeeperCut (Insafutdinov et al., 2016), OpenPose (Cao et al., 2017), Associative Embedding (Newell et al., 2017), PersonLab (Papandreou et al., 2018), PifPaf (Kreiss et al., 2019) and CenterNet (Zhou et al., 2019), keypoint detection and grouping are usually regarded as two heterogeneous learning targets. This requires the model to learn the keypoint heatmaps encoding position information and the human knowledge guided signals encoding association information such as part hypotheses, part affinity fields, associative embeddings or offset vector fields.

In this paper we explore whether we can exploit the instance semantic clues implicitly used by the model to group the detected keypoints into individual instances. Our key intuition is that, when the model predicts a location of a specific keypoint, it may know the human instance region this keypoint belongs to, which means that the model has implicitly associated related joints together. For example, when an elbow is recognized, the model may learn its strong spatial dependencies in its adjacent wrist or shoulder but weak dependencies in the joints of other persons. Therefore, if we can read out such information learned and encoded in the model, the detected keypoints can be correctly grouped into instances, without the help of the human pre-defined associative signals.

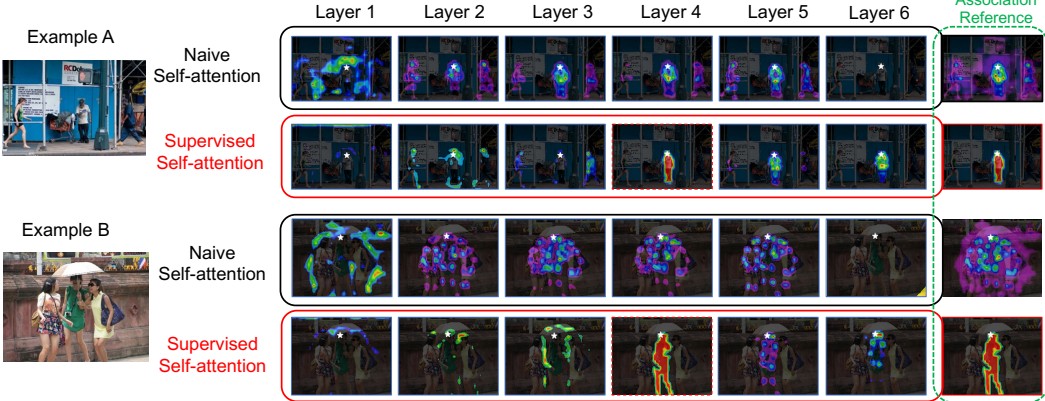

Figure 1: We choose two examples to show differences between the naive and supervised self-attention patterns. The association reference for the naive self-attention is the average of all layers.

We argue that self-attention model (Vaswani et al., 2017) can meet this requirement because it can provide image-specific pairwise similarities between any pair of image positions without distance limitation, and the resulting attention patterns show object-related semantics. Hence, we attempt to use self-attention mechanism to perform multi-person pose estimation. But instead of following the top-down strategy with the single person region as the input, we feed the Transformer model the high-resolution input images with the presence of multiple persons, and expect it to output the heatmaps encoding multi-person keypoint locations. Our initial results show that 1) the heatmaps outputted by Transformer can also accurately respond to multiple persons' keypoints at multiple candidate locations; 2) the attention scores between the detected keypoint locations tend to be higher within the same person but lower across different persons. Based on these findings, we introduce an attention-based parsing algorithm to group the detected keypoints into different human instances.

Unfortunately, the naive self-attention does not always show desirable properties. In many cases, a detected keypoint also probably have relatively higher attention scores with those belonging to different person instances. This will definitely lead to wrong associations and implausible human poses. To address this issue, we propose a novel method that leverages a loss function to explicitly supervise the attention area of each person instance by the mask of the instance. The results show that supervising self-attentions in such a way can achieve the expected instance-discriminative characteristics without affecting the standard forward propagation of Transformer. Such characteristics guarantee the effectiveness and accuracy of the attention-based grouping algorithm. The results on the COCO keypoint detection challenge show that our models with limited refinement can achieve comparable performances compared with the highly optimized bottom-up pose estimation systems (Cao et al., 2017; Newell et al., 2017; Papandreou et al., 2018). Meanwhile, we also can easily obtain the person instance masks by sampling the corresponding attention areas, thereby avoiding an extra pixel assignment or grouping algorithm.

## 1.1 KEY CONTRIBUTIONS

**Using self-attention to unify keypoint detection, grouping and human mask prediction**. We use Transformer to solve the challenging multi-person keypoint detection, grouping and mask prediction in a unified way. We realize that the self-attention shows instance-related semantics, which can be served as the association information in a bottom-up fashion. We further use instance masks to supervise the self-attention. It ensures that each keypoint is assigned to the correct human instance according to the attention scores, making it easy to obtain the instance masks as well.

**Supervising self-attention "for your need"**. A common practice of using Transformer models is to use task-specific signals to supervise the final output of transformer-based models, such as class labels, object box coordinates, keypoint positions or semantic masks. In this method, a key novelty is to use some type of constraint terms to control the behaviors of self-attention. The results show that under supervision the self-attention can achieve instance-aware characteristics for multi-person

pose estimation and mask prediction, without destroying the standard forward of Transformer. This demonstrates that using appropriate guidance signals makes self-attention controllable and help the model learning, which is also applicable to other vision tasks such as instance segmentation (Wang et al., 2021) and object detection (Carion et al., 2020).

## 2 METHOD

### 2.1 PROBLEM SETTING

Given a RGB image $I$ of size $3 \times H \times W$, the goal of 2D multi-person pose estimation is to estimate all persons' keypoints locations: $\mathbb{S} = \left\{ (x_i^k, y_i^k) | i = 1, 2, ..., N; k = 1, 2, ..., K \right\}$, where $N$ is the number of persons in this image and $K$ is the number of defined keypoint types.

We follow the bottom-up strategy. First, the model detects all the candidate locations for each type of keypoints in an image: $\mathbb{C} = \mathbb{C}_1 \bigcup \mathbb{C}_2 \bigcup ... \bigcup \mathbb{C}_K$, where $\mathbb{C}_k = \{ (\hat{x}_i, \hat{y}_i) | i = 1, 2, ..., N_k \}$ represents the $k$-th type of keypoint set with $N_k$ detected candidates. Second, a heuristic decoding algorithm $g$ groups all candidates into $M$ skeletons based on the association information $\mathcal{A}$, which determines a unique person ID for each keypoint location. We formulate this process as: $g((\hat{x}_i, \hat{y}_i), \mathbb{C}, \mathcal{A}) \to m \in \{1, 2, ..., M\}$.

Next, we present the model architecture and show how to use self-attention as the association information $\mathcal{A}$. We analyze the problems when using the naive self-attention as the grouping reference. We propose to supervise self-attention via instance masks for keypoints grouping. We present two types of grouping algorithm from the *body-first* and *part-first* views. Finally, we describe how we obtain the person instance masks and how we use the obtained masks to refine the results.

### 2.2 NETWORK ARCHITECTURE AND NAIVE SELF-ATTENTION

**Architecture.** We use a simple architecture combination that includes ResNet (He et al., 2016) and Transformer encoder (Vaswani et al., 2017), like the design of TransPose (Yang et al., 2021). The downsampled feature maps of ResNet with $r$ stride are flattened to a sequence of $L \times d$ size and sent to Transformer where $L = \frac{H}{r} \times \frac{W}{r}$. Several transposed convolutions and a $1 \times 1$ convolution are used to upsample the Transformer output into the target keypoint heatmap size $K \times \frac{H}{4} \times \frac{W}{4}$.

**Heatmap loss.** To observe what patterns the self-attentions layers spontaneously learn, we first only leverage the mean square error (MSE) loss between the predicted heatmap $\hat{\mathbf{H}}_k$ and the groundtruth heatmap $\mathbf{H}_k$ to train the model:

$$\mathcal{L}_{heatmap} = \frac{1}{K} \sum_{k=1}^{K} \mathbf{M} \cdot \left\| \hat{\mathbf{H}}_k - \mathbf{H}_k \right\|, \qquad (1)$$

where $\mathbf{M}$ is a mask that masks out the crowd areas and small size person segments in the whole image. After the model is trained only by heatmap loss, the keypoint detection results show the trained model can accurately localize keypoints of multiple persons.

**Issues in naive self-attention.** We obtain the keypoint locations from heatmaps and further visualize the attention areas of these locations. As revealed by the examples shown in Figure 1, using the naive self-attention matrices as the association reference poses several challenges: 1) There are multiple attention layers in Transformer, each of which shows distinct characteristics. Selecting which attention layers as the association reference and how to process the raw attention require a very thoughtful fusion and post-processing strategy. 2) Although most of the sampled keypoint locations show local attention areas, especially for the people they belong to, some keypoints may still produce relatively high attention scores for the parts of other people at a longer distance. It is almost impossible to determine a perfect attention threshold for all situations, which makes keypoint grouping highly dependent on specific experimental observations. As a consequence, the attention-based grouping cannot ensure the correctness of the keypoint assignment, leading to inferior performance.

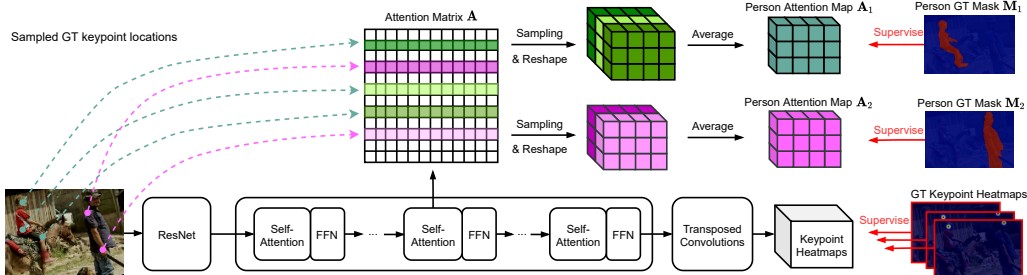

Figure 2: Model overview. The model architecture consists of three parts: a regular ResNet, a regular Transformer encoder, and several transposed convolutional layers. Two types of loss function are leveraged to supervise the model training. The final output of the model is supervised by the groundtruth keypoint heatmaps. One of the immediate self-attention layers is sparsely supervised by the instance masks. In particular, we sample the rows of the attention matrix of the chosen attention layer according to the visible keypoint locations of each human instance, reshape them into 2D-like maps, and then use the mask of each instance to supervise the average map. In this figure, we only show a few keypoints of each instance for simplicity.

## 2.3 SUPERVISING SELF-ATTENTION BY INSTANCE MASKS

To address the aforementioned challenges of using the naive self-attention for keypoints grouping, we **S**upervise **S**elf-**A**ttention (**SSA**) to be what we expect. Ideally, the expected attention pattern should be that each keypoint location only attends to the person instance it belongs to. The value distribution (0 or 1) in a person instance mask provides an ideal guidance signal to supervise the pairwise keypoints's locations to have lower or higher attention scores. Then we propose a sparse sampling method based on the instance keypoint locations to supervise the specific attention matrix generated by the self-attention computation in Transformer, as illustrated in Figure 2.

**Instance mask loss.** We suppose that the $p$-th person's keypoints groudtruth locations are $\left\{(x_p^k, y_p^k, v_p^k)\right\}_{k=1}^{K}$, where $v_p^k \in \{0, 1\}$ is a visibility flag, i.e., $v_p^k = 0$: not labeled, $v_p^k = 1$: labeled. We take out the immediate attention matrix $\mathbf{A} = \text{Softmax}(\frac{\mathbf{Q}\mathbf{K}^\top}{\sqrt{d}}) \in \mathbb{R}^{L \times L}$ of the specific layer in Transformer[1] to leverage the supervision. We first reshape the attention matrix $\mathbf{A}$ into a tensor $\mathbf{A}$ of $(h \times w) \times (h \times w)$ size, where $h = H/r, w = W/r$. Then we transform the keypoint coordinates into the coordinate system of the downsampled feature maps. And then we take out the corresponding rows of the attention matrix specified by these locations. So we can obtain the reshaped attention map at each keypoint location: $\mathbf{A}[int(y_p^k/r), int(x_p^k/r), :, :]$. For a person instance, we sample and average the attention maps based on its *visible* keypoint locations to estimate the mean attention map. We name it as *person attention map* $\mathbf{A}_p$:

$$\mathbf{A}_p = \frac{1}{\sum_{i=1}^{K} v_p^i} \sum_{k=1}^{K} v_p^k \cdot \mathbf{A}[int(y_p^k/r), int(x_p^k/r), :, :]. \tag{2}$$

Assuming the groundtruth mask of the $p$-th person in the image is $\mathbf{M}_p$, we also use the MSE loss function to supervise the attention matrix sparsely. Since the self-attention scores have been normalized by the softmax function, we need to rescale the $\mathbf{A}_p$ by dividing its maximum value so that the rescaled $\mathbf{A}_p$ is closer to the value range of the annotated mask. Note that the size of $\mathbf{A}_p$ is $\frac{H}{r} \times \frac{W}{r}$ while the groundtruth instance mask is constructed to be $\frac{H}{4} \times \frac{W}{4}$ size. So we use $r/4$ times bilinear interpolation to resize the $\mathbf{A}_p$ to have the same size as the instance mask. We formulate the instance mask loss as:

$$\mathcal{L}_{mask} = \text{MSE}(\text{bilinear}(\mathbf{A}_p/\max(\mathbf{A}_p)), \mathbf{M}_p) = \frac{1}{N}\sum_{p=1}^{N} \|\text{bilinear}(\mathbf{A}_p/\max(\mathbf{A}_p)) - \mathbf{M}_p\|. \tag{3}$$

---

[1]$\mathbf{Q}, \mathbf{K} \in \mathbb{R}^{L \times d}$ are queries and keys. For simplicity we consider there is only one head. For multihead self attention, the attention matrix $\mathbf{A}$ is the average of all heads' attention matrices.

**Objective.** So the overall objective for training the model is:

$$\mathcal{L}_{train} = \alpha \cdot \mathcal{L}_{heatmap} + \beta \cdot \mathcal{L}_{mask}, \tag{4}$$

where $\alpha$ and $\beta$ are two coefficients to balance two types of learning. In the standard self-attention computation of Transformer, the attention matrix is computed by the inner products of *queries* and *keys*. Its gradient back-propagation information is entirely derived from the subsequent attention weighted sum of *values*. By introducing the instance mask loss to supervise the self-attention, the gradient learning direction for the supervised attention matrix has two sources: the implicit gradient signal from keypoint heatmaps learning and the explicit similarity constraint from instance mask learning. Choosing approximate values of $\alpha$ and $\beta$ is critical for training the model well. We set $\alpha = 1, \beta = 0.01$ to balance both heatmap learning and mask learning.

## 2.4 KEYPOINTS GROUPING

When the well-trained model makes a single forward pass for a given image, we can decode the multi-person human poses and masks from the outputted keypoint heatmaps and the supervised attention matrix in the immediate attention layer. We first conduct non-maximum suppression in a 7×7 local window on the keypoint heatmaps and obtain all local maximum locations whose scores exceed the threshold $t$. We put all these candidates into a queue and decode them into skeletons using the attention-based algorithm. Using the self-attention similarity matrix with quadratic complexity in-

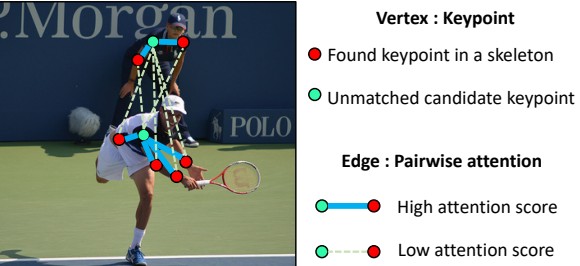

Figure 3: Self-Attention based Grouping. When the founded keypoints in a skeleton induce a stronger attention attraction to an unmatched keypoint, this candidate will be assigned to this skeleton. The blue edges (thick) have totally higher attention scores than the green edges (slim).

evitably brings redundant computation. However, in part, this also makes minimal assumptions about where the keypoints of the instances may appear and the number of persons in the image. Next we present the self-attention based algorithms from the *body-first* and *part-first* views.

**Body-first view**. This view aims to decode each person skeleton one-by-one from the queue. Assuming we have the sorted all types of candidate keypoints by descending order of score in a single queue, we pop out the first keypoint (maybe any keypoint type) to seed a new skeleton $\mathcal{S}$, and then greedily find the best matched adjacent candidate keypoint from the queue.

For the seeded $\mathcal{S}$ with the initial keypoint, we find the other keypoints along the search path according to a defined human skeleton kinematic tree. When looking for a certain type of joint, the founded joints (denoted as the set $\mathcal{S}_f$) of this skeleton $\mathcal{S}$ induce a basin of attraction to "attract" the joint that most likely belongs to it, as illustrated in Figure 3. For a certain unmatched point $p_c = (x, y, s)$ in the candidate set $\mathbb{C}_k$ of the keypoint type $k$, we use the mean attention scores between the current found keypoints and $p_c$ as the metric to measure the attraction from this skeleton[2]: $\text{Attraction}(p_c, \mathcal{S}_f) = \frac{1}{|\mathcal{S}_f|} \sum_{(x',y',s')\in\mathcal{S}_f} s' \cdot \mathbf{A}[y, x, y', x']$. Thus the candidate point with the highest score × Attraction is considered to belong to the current skeleton $\mathcal{S}$:

$$p_c^* = \text{argmax}_{p_c \in \mathbb{C}_k} s \cdot \text{Attraction}(p_c, \mathcal{S}_f). \tag{5}$$

We repeat the process above and record all the matched keypoints until all keypoints of this skeleton have been found. Then we need to decode the next skeleton. We pop the first unmatched keypoint to seed a new skeleton $\mathcal{S}'$ again. We follow the previous steps to find keypoints belonging to this instance. Note if the $\text{Attraction}(p_c^*, \mathcal{S}_f)$ is smaller than a threshold $\lambda$ (empirically set to 0.0025), this type of keypoint in this skeleton to be empty (zero-filling). It is also worth noting that we also consider the keypoints that have already been claimed by a previous skeleton $\mathcal{S}$, but only when $\text{Attraction}(p_c, \mathcal{S}_f') > \text{Attraction}(p_c, \mathcal{S})$, we assign the matched $p_c$ to the current skeleton $\mathcal{S}'$.

---

[2]To obtain the correct coordinate $(int(x/r), int(y/r))$, we use $(x, y)$ to omit the downsampling factor and rounding operation for simplicity.

Table 1: Results on the COCO validation set. (res101, s16, i640) represents that we use ResNet-101; the output stride is 16; the input resolution is 640×640. Refinement#1 represents only refining the keypoints without filling. Refinement#2 represents refining the keypoints with filling.

| Method | AP | $AP_{0.5}$ | $AP_{0.75}$ | $AP_M$ | $AP_L$ | AR |
|---|---|---|---|---|---|---|
| OpenPose (Cao et al., 2017) | 58.4 | 81.5 | 62.6 | 54.4 | 65.1 | - |
| OpenPose + Refinement (Cao et al., 2017) | 61.0 | 84.9 | 67.5 | 56.3 | 69.3 | - |
| PersonLab (res101, s16, i601) (Papandreou et al., 2018) | 53.2 | 76.0 | 56.3 | 38.6 | 73.1 | 57.0 |
| PersonLab (res101, s16, i801) (Papandreou et al., 2018) | 60.0 | 82.1 | 64.3 | 49.7 | 74.6 | 64.1 |
| PersonLab (res101, s16, i1401) (Papandreou et al., 2018) | 65.6 | 85.9 | 71.4 | 61.1 | 72.8 | 70.1 |
| Ours (res101, s16, i640) | 50.4 | 78.5 | 53.1 | 41.6 | 62.8 | 56.9 |
| Ours (res152, s16, i640) | 50.7 | 77.7 | 53.6 | 41.1 | 64.2 | 56.9 |
| Ours (res152, s16, i640) + Refinement#1 | 58.7 | 81.1 | 62.9 | 54.0 | 66.0 | 63.9 |
| Ours (res152, s16, i640) + Refinement#2 | 65.3 | 85.8 | 71.3 | 59.1 | 74.4 | 70.5 |
| Ours (res101, s16, i800) | 51.6 | 79.7 | 55.1 | 44.6 | 61.2 | 57.9 |
| Ours (res101, s16, i800) + Refinement#1 | 59.3 | 82.1 | 63.7 | 56.4 | 63.6 | 64.6 |
| Ours (res101, s16, i800) + Refinement#2 | 66.4 | 86.1 | 72.6 | 61.1 | 74.0 | 71.2 |

**Part-first view**. This view aims to decode all human skeletons part-by-part. Given all candidates for each keypoint type, we initialize multiple skeleton seeds $\{\mathcal{S}^1, \mathcal{S}^2, ..., \mathcal{S}^m\}$ with the most easily detected keypoints such as nose. Then we follow a fixed order to connect the candidate parts to the current skeletons. These skeletons can be seen as multiple clusters consisting of found keypoints. Like the *body-first* view, we also use the mean attention attraction $\text{Attraction}(p_c, \mathcal{S}_f^t)$ from the found keypoints in the skeletons as the metric to assign the candidate parts (Figure 3). But in the *part-first* view, we compute the pairwise distance matrix between the candidate parts and existing skeletons, and then we use the Hungarian algorithm (Kuhn, 1955) to solve this bipartite graph matching problem. Note, if an $\text{Attraction}(p_c, \mathcal{S}^t)$ that represents a matching in the solution is lower than a threshold $\lambda$, we use this corresponding candidate part to start a new skeleton seed. We repeat the process above until all types of candidate parts have been assigned. This part-first grouping algorithm can achieve the optimal solution for assigning local parts to the skeletons although it cannot guarantee the global optimal assignment. We choose the part-first grouping as the default. And we compare both algorithms on the performance, complexity and runtime in Appendix A.5.

## 2.5    MASK PREDICTION

The instance masks are easy to obtain after the detected keypoints have been grouped into skeletons. To produce the instance segmentation results, we sample the visible keypoint locations $\left\{(\hat{x}_m^k, \hat{y}_m^k, \hat{v}_m^k)\right\}_{k=1}^K$ of the $m$-th instance from the supervised self-attention matrix: $\hat{\mathbf{A}}_m = \frac{\sum_k \delta(\hat{v}_m^k > 0) \cdot \mathbf{A}[\hat{y}_m^k, \hat{x}_m^k, :, :]}{\sum_k \delta(\hat{v}_m^k > 0)}$. Then we achieve the estimated instance mask: $\hat{\mathbf{M}}_m = \frac{\hat{\mathbf{A}}_m}{\max(\hat{\mathbf{A}}_m)} > \sigma$, where $\sigma$ is a threshold (0.4 by default) to determine the mask region. When we obtain the initial skeletons and masks for all person instances, the joints of a person may fall in multiple incomplete skeletons, but their corresponding segments (sampled attention areas) may overlap. Thus we further perform non-maximum suppression to merge instances if the Intersection-over-Max (IoM) of two masks exceeds 0.3, where Max denotes the maximum area between two masks.

## 3    EXPERIMENTS

**Dataset**. We evaluate our method on the COCO keypoint dectection challenge (Lin et al., 2014) and on the instance segmentation of the COCO person category.

**Model setup**. We follow the model architecture design of TransPose (Yang et al., 2021) to predict the keypoint heatmaps. The setup is built on top of pre-existing ResNet and Transformer Encoder. We use the Imagenet pre-trained ResNet-101 or ResNet-151 as the backbone whose final classification layer is replaced by a $1 \times 1$ convolution to reduce the channels from 2048 to $d$ (192). The normal output stride of ResNet backbone is 32 but we increase the feature map resolution of its final stage (C5 stage) by adding the dilation and removing the stride, i.e., the downsampling ratio $r$ of ResNet

is 16. We use a regular Transformer with 6 encoder layers with a single attention head for each layer. The hidden dimension of FFN is 384. See more training and inference details in Appendix A.1.

## 3.1 RESULTS ON COCO KEYPOINT DETECTION AND PERSON INSTANCE SEGMENTATION

The standard evaluation metric for COCO keypoint localization is the object keypoint similarity (OKS) and the mean average precision (AP) over 10 thresholds (0.5,0.55,...,0.95) is regarded as the performance metric. We train our models on COCO train2017 set, and evaluate the model on the val2017 and test-dev2017 sets, as shown in Table 1 and Table 2. We mainly compare with the typical bottom-up models that have similar pipelines to our method: OpenPose (Cao et al., 2017), PersonLab (Papandreou et al., 2018), and AE (Newell et al., 2017). Following the works (Cao et al., 2017; Newell et al., 2017), we also refine the grouped skeletons using a single pose estimator. We adopt the COCO pretrained TransPose-R-A4 (Yang et al., 2021) that has a very similar architecture to our model and has only 6M parameters. We apply the single pose estimator to each single scaled person region achieved by the box containing the person mask. Note that the refinement results are highly dependent on the effect of the grouping and mask prediction, and we only update the keypoint estimates where the predictions of the two models are almost the same. The concrete update rule is whether the keypoint similarity (KS) metric[3] computing between two keypoints exceeds 0.75, indicating that the distance between two predicted locations is already very small.

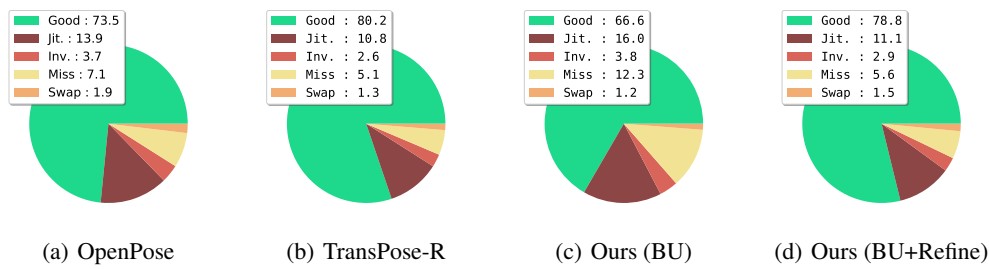

(a) OpenPose      (b) TransPose-R      (c) Ours (BU)      (d) Ours (BU+Refine)

Figure 4: Localization errors analysis on COCO validation set.

**Analysis**. We further analyze the differences between pure bottom-up results and the refined ones through the benchmarking and error diagnosis tool (Ronchi & Perona, 2017). We compare our methods with the typical OpenPose model and the Transformer-based model. The yielded localization error bars (Figure 4) reveal the weaknesses and strengths of our model: (1) *Jitter error*: The heatmap localization precision under the existence of multi-person is still not as accurate as the localization precision of single pose estimate; (2) *Missing error*: Since our algorithm does not ensure that the coordinate of every keypoint in a detected pose has been predicted, if the GT coordinate of a keypoint is annotated, zero-filling coordinates will seriously pull down the calculated OKS value. Thus, for the evaluation, it is necessary to produce complete predictions. When we further use the single pose estimator to fill the missing joints with zero scores in the initially grouped skeletons, it achieves about 7 AP gains (Table 1) and reduces the missing error (shown in Figure 4(d)); (3) *Inversion error*: Forcing diverse keypoint types in an individual instance to have higher query-key similarity may make it difficult for the model to distinguish different keypoint types, especially the left and right inversion; (4) *Swap error*: We notice that our pure bottom-up model has fewer swap errors (1.2%, shown in Figure 4(c)), which represents less confusion between semantically similar parts of different instances. It indicates that compared with OpenPose model, our attention-based grouping strategy performs relatively better in assigning parts to their corresponding instances. We show the qualitative human poses and instance segmentation results in Appendix A.6.

**Person instance segmentation**. We evaluate the instance segmentation results on COCO val split (person category only). We compare our method with PersonLab (Papandreou et al., 2018). In Table 3, we report the results with a maximum of 20 person proposals due to the convention of the COCO person keypoint evaluation protocol. The results on the mean average precision (AP) show that our model still has a gap in the segmentation performance compared with PersonLab. We argue that this is mainly because we conduct the mask learning on low-resolution attention maps that have

---

[3]We consider the per-keypoint standard deviation and object scale as the standard OKS metric does.

Table 2: Results on the COCO test-dev2017 set. The methods marked with $*$ use the multi-scale inference settings. Our result is achieved based on ResNet-101 model with 16 output stride and $800^2$ input resolution.

| Method | AP | $AP_{0.5}$ | $AP_{0.75}$ | $AP_M$ | $AP_L$ | AR | $AR_{0.5}$ | $AR_{0.75}$ | $AR_M$ | $AR_L$ |
|---|---|---|---|---|---|---|---|---|---|---|
| Top-down | | | | | | | | | | |
| G-RMI (Papandreou et al., 2017) | 64.9 | 85.5 | 71.3 | 62.3 | 70.0 | 69.7 | 88.7 | 75.5 | 64.4 | 77.1 |
| Mask-RCNN (He et al., 2017) | 63.1 | 87.3 | 68.7 | 57.8 | 71.4 | - | - | - | - | - |
| SimpleBaseline (Xiao et al., 2018) | 73.7 | 91.9 | 81.1 | 70.3 | 80.8 | 79.0 | - | - | - | - |
| HRNet (Sun et al., 2019) | 75.5 | 92.5 | 83.3 | 71.9 | 81.5 | 80.5 | - | - | - | - |
| Bottom-up | | | | | | | | | | |
| OpenPose (Cao et al., 2017) | 61.8 | 84.9 | 67.5 | 57.1 | 68.2 | - | - | - | - | - |
| AE (Newell et al., 2017) | 62.8 | 84.6 | 69.2 | 57.5 | 70.6 | - | - | - | - | - |
| AE$^*$ (Newell et al., 2017) | 65.5 | 86.8 | 72.3 | 60.6 | 72.6 | 70.2 | 89.5 | 76.0 | 64.6 | 78.1 |
| PersonLab (Papandreou et al., 2018) | 66.5 | 88.0 | 72.6 | 62.4 | 72.3 | 71.0 | 90.3 | 76.6 | 66.1 | 77.7 |
| PersonLab$^*$ (Papandreou et al., 2018) | 68.7 | 89.0 | 75.4 | 64.1 | 75.5 | 75.4 | 92.7 | 81.2 | 69.7 | 83.0 |
| SPM (Nie et al., 2019) | 66.9 | 88.5 | 72.9 | 62.6 | 73.1 | - | - | - | - | - |
| HigherHRNet$^*$ (Cheng et al., 2020) | 70.5 | 89.3 | 77.2 | 66.6 | 75.8 | - | - | - | - | - |
| DEKR$^*$ (Geng et al., 2021) | 71.0 | 89.2 | 78.0 | 67.1 | 76.9 | 76.7 | 93.2 | 83.0 | 71.5 | 83.9 |
| **Ours** (SSA) | 65.0 | 86.2 | 72.2 | 60.1 | 71.8 | 70.1 | 88.9 | 76.2 | 64.2 | 78.2 |

Table 3: Instance segmentation results (person class only) obtained with 20 proposals per image on the COCO validation set.

| Method | AP | $AP_{0.5}$ | $AP_{0.75}$ | $AP_{small}$ | $AP_{medium}$ | $AP_{large}$ | $AR_1$ | $AR_{10}$ | $AR_{20}$ | $AR_{small}$ | $AR_{medium}$ | $AR_{large}$ |
|---|---|---|---|---|---|---|---|---|---|---|---|---|
| PersonLab (res101, stride=8, input=1401) | 33.8 | 56.0 | 36.8 | 7.6 | 45.9 | 59.1 | 15.6 | 37.0 | 38.3 | 8.0 | 51.4 | 68.0 |
| Ours (res152, stride=16, input=640) | 20.7 | 43.5 | 16.9 | 0.3 | 24.5 | 59.0 | 12.9 | 29.4 | 30.3 | 1.0 | 36.1 | 68.5 |
| Ours (res101, stride=16, input=800) | 22.0 | 45.3 | 18.8 | 0.9 | 27.7 | 55.3 | 13.2 | 30.8 | 32.0 | 1.8 | 41.1 | 66.9 |

been downsampled 16 times w.r.t. the $640^2$ or $800^2$ input resolution while the reported PersonLab result is based on 8 times downsampling w.r.t the $1401^2$ input resolution. As shown in Table 3, our model performs worse on small and medium scales but achieves comparable or even superior performance on large scale persons even if PersonLab uses a larger resolution.

## 3.2 COMPARISON BETWEEN NAIVE SELF-ATTENTION AND SUPERVISED SELF-ATTENTION

To study the differences in model learning when trained with and without supervising self-attention, we compare their convergences in the heatmap loss and instance mask loss, since the overfitting on COCO train data is usually not an issue. As illustrated in Figure 5, compared with training the naive self-attention model, supervising self-attention achieves a better fitting effect in the mask learning, while achieving an acceptable sacrifice on the fitting of heatmap learning. It is worth noting that the instance mask training loss curve of the naive self-attention model drops slightly, which suggests that the spontaneously formed attention pattern has a tendency to instance-awareness. To quantitatively evaluate the performance of using naive self-attention patterns for keypoint grouping, we average the attentions from all transformer layers as the association reference (shown in Figure 1). When we use the totally same conditions (including model configuration, training  testing settings and grouping algorithm) of the supervised self-attention model based on (res152, s16, i640), we achieve 29.0AP on COCO validation set, which is far from the 50.7AP result achieved by supervising self-attention.

## 4 RELATED WORK

**Transformer.** Transformer (Vaswani et al., 2017) has shown very powerful visual relation modeling capability in various computer vision tasks, such as image classification (Dosovitskiy et al., 2020; Touvron et al., 2020), object detection (Carion et al., 2020; Zhu et al., 2020), semantic segmentation (Zheng et al., 2021), tracking (Sun et al., 2020; Meinhardt et al., 2021), human pose estimation (Lin et al., 2021; Li et al., 2021a; Yang et al., 2021; Li et al., 2021b; Stoffl et al., 2021) and etc. The common practice of these methods is to use the task-specific supervision signals such as class labels, object box coordinates, keypoint positions or semantic masks to supervise the final output of transformer-based models. They may visualize the attention maps to understand the

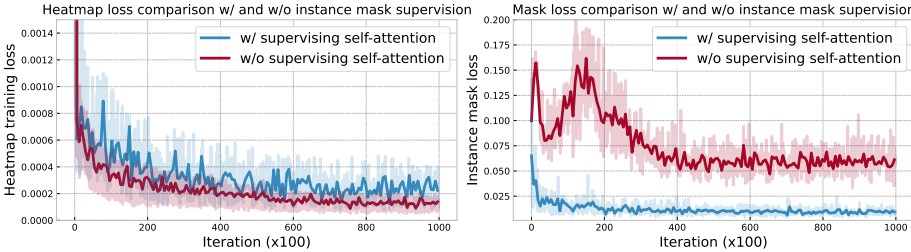

Figure 5: The convergences on the heatmap loss and instance mask loss when trained with and without supervising self-attention.

model but few works directly use it as an explicit function in the inference process. Our work gives a successful example of explicitly using and supervising attention for a specific purpose.

**Human Pose Estimation & Instance segmentation.** Multi-person pose estimation methods are usually classified into two categories: top-down (TD) or bottom-up (BU). TD models first detect persons, and then estimate single pose for each person, such as G-RMI (Papandreou et al., 2017), Mask-RCNN (He et al., 2017), CPN (Chen et al., 2018), SimpleBaseline (Xiao et al., 2018), and HRNet (Sun et al., 2019). BU models need to detect the existence of various types of keypoints at any position and scale. And matching keypoints into instances requires the model to learn dense association signals pre-defined by human knowledge. OpenPose (Cao et al., 2017) proposes part affinity field (PAF) to measure the association between keypoints by computing the integral along the connecting line. Associative Embedding (Newell et al., 2017) abstracts an embedding as the human 'tag' ID to measure the association. PersonLab (Papandreou et al., 2018) constructs mid-range offset as the geometric embedding to group keypoints into instances. In addition, single-stage methods (Zhou et al., 2019; Nie et al., 2019) also regress offset field to assign keypoints to their centers. Compared with them, we use Transformer to capture the intra-dependencies within a person and inter-dependencies across different persons. And we explicitly exploit the intrinsic property of self-attention mechanism to solve the association problem, rather than regressing highly abstracted offset fields or embeddings. The generic instance segmentation methods also can be categorized into top-down and bottom-up schemes. Top-down approaches predict the instance masks based on the object proposals, such as FCIS (Li et al., 2017) and Mask-RCNN (He et al., 2017). Bottom-up approaches mainly cluster the semantic segmentation results to obtain instance segmentation using an embedding space or a discriminative loss to measure the pixel association like (Newell et al., 2017; De Brabandere et al., 2017; Fathi et al., 2017). Compared with them, our method uses self-attention to measure the association and estimates instance masks based on instance keypoints.

## 5 DISCUSSION AND FUTURE WORKS

This paper presents a new method to solve keypoint detection and instance association by using Transformer. We supervise the inherent characteristics of self-attention – the feature similarity between any pair of positions – to solve the grouping problem of the keypoints or pixels. Unlike a typical CNN-based bottom-up model, it no longer requires a pre-defined vector field or embedding as the associative reference, thus reducing the model redundancy and simplifying the pipeline. We demonstrate the effectiveness and simplicity of the proposed method on the challenging COCO keypoint detection and person instance segmentation tasks.

The current approach also brings limitations and challenges. Due to the quadratic complexity of the standard Transformer, the model still struggles in simultaneously scaling up the Transformer capacity and the resolution of the input image. The selection of loss criteria, model architecture, and training procedures can be further optimized. In addition, the reliance on the instance mask annotations also can be removed in future works, such as by imposing high and low attention constraints only on the pairs of keypoint locations. While, the current approach still has not yet beaten the CNN-based bottom-up state-of-the-art counterparts developed with many sophisticated designs, we believe it is promising to exploit or supervise self-attention to solve the detection and association problems in multi-person pose estimation, and other tasks or applications.

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

# A   APPENDIX

## A.1   TRAINING DETAILS

In the training phase, we use data augmentation with random scale factor between 0.75 and 1.5, random flip with probability 0.5, random rotation with $\pm 30$ degrees, and random translate with $\pm 40$ pixels along the horizontal and vertical directions. The input size is $640^2$ or $800^2$ and thus the input sequence length of Transformer is 1600 or 2500. We use the Post-Norm Transformer architecture and ReLU activation function in FFN. We supervise the 4-th self-attention layer by default (the total layer depth is 6). By convention, we use 2 transposed convolution layers to upsample the Transformer output size to $160 \times 160$ or $200 \times 200$. The standard deviation for the Gaussian kernel in the generated heatmaps is set to 2. We use Adam optimizer to train the model. The model is distributed across 8 Tesla V100 GPUs with a total batchsize of 16 or 8. The initial learning rate is set to $\frac{\text{batchsize}}{8} \times 0.0001$, and decays 10 times at the 150-th and 200-th epochs respectively, with a total of 240 training epochs.

## A.2   INFERENCE DETAILS

The threshold score $t$ for obtaining candidate keypoint from heatmaps is set to 0.0025. The final person masks are achieved by bilinear interpolating the estimated $\hat{\mathbf{M}}_m$ to the original image size. The skeleton kinematic tree used in the body-first grouping is defined as a graph structure: the vertices are all types of keypoints that are denoted as the numbers from 0 to 16 by the order defined by COCO dataset; the edges are defined as [(0, 1), (0, 2), (0, 3), (0, 4), (3, 5), (4, 6), (5, 7), (5, 11), (6, 8), (6, 12), (7, 9), (8, 10), (11, 13), (13, 15), (12, 14), (14, 16), (5, 6), (15, 16), (13, 14), (11, 12)].

## A.3   ABLATION ON WHICH ATTENTION LAYERS SHOULD BE SUPERVISED

Supervising the self-attention matrix in different Transformer layer depths may have different effects on the heatmap and mask learning. To study such effects, we train a smaller proxy model to compare their differences in the fitting of heatmap and instance mask loss on a small subset (1/5) split of the COCO train set. The model configurations are: ResNet-50 based, $576^2$ input resolution and 5 transformer layers with $d = 160$ and 320 hidden dimensions in FFN. Note that using a smaller model and small-scale training data inevitably reduces the overall performances of the model, but we only aim to find the relative differences in supervising at different Transformer layers. As illustrated in Figure 6, we do not observe significant differences in both heatmap loss and instance mask loss when leveraging the mask supervision in different Transformer layers. We further evaluate all these models on the COCO validation set. As shown in Table 4, supervising one of the last three attention layers achieves better performance compared with supervising the first two layers. Especially, supervising the penultimate or third-to-last layer shows a better performance. This suggests that leveraging the instance mask loss in this layer depth is a better trade-off between heatmap learning and mask learning.

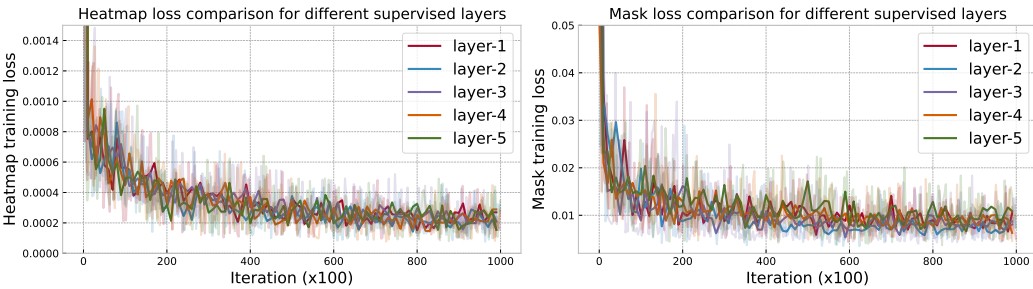

Figure 6: The convergences on the heatmap loss and mask loss when supervising the self-attention in different layer depths.

| Supervised layer | AP | $AP_{0.5}$ | $AP_{0.75}$ | $AP_M$ | $AP_L$ | AR | AP (with Refinement) |
|---|---|---|---|---|---|---|---|
| 1-th | 32.3 | 60.9 | 29.8 | 23.0 | 45.5 | 38.8 | 52.1 |
| 2-th | 33.7 | 62.1 | 31.7 | 22.9 | 48.8 | 40.0 | 54.6 |
| 3-th | 34.1 | 63.0 | 31.4 | 23.4 | 49.0 | 40.4 | 54.6 |
| 4-th | 34.1 | 63.0 | 32.0 | 23.3 | 49.0 | 40.2 | 54.7 |
| 5-th | 33.9 | 62.7 | 31.4 | 23.5 | 48.5 | 40.5 | 54.7 |

Table 4: Comparisons for different supervised layers on COCO validation set when using a small proxy model.

### A.4 WILL AN INDEPENDENT SELF-ATTENTION HEAD BE BETTER THAN A SHARED ONE TO LEVERAGE THE INSTANCE MASK LOSS?

Intuitively, using an independent self-attention head may be helpful to reduce the effect of introducing an intermediate instance mask loss on the standard Transformer forward. Thus we try to mitigate the negative effect on the heatmap localization by using an independent self-attention head to leverage the mask supervision. This design will need to insert an extra self-attention layer to the transformer intermediate output, as shown in Figure 7. However, by comparing the convergence of the training losses, we find no obvious difference in the heatmap loss fitting between using shared self-attention attention and independent self-attention, while, the independent self-attention performs relatively better in fitting the instance mask loss.

When we test their performances on COCO validation set, we find both designs achieve similar performances, as shown in Table 5. Such results indicate that using an independent layer to the intermediate loss bring little gain, and introducing an intermediate instance mask loss may generate a weak effect on the prediction of keypoint heatmaps. We conjecture that the existence of the residual path parallel to the supervised self-attention layer may also adaptively reduce the effect of the instance mask loss on the subsequent transformer layers, since we only leverage the sparse constraints to the self-attention matrix in a certain transformer layer.

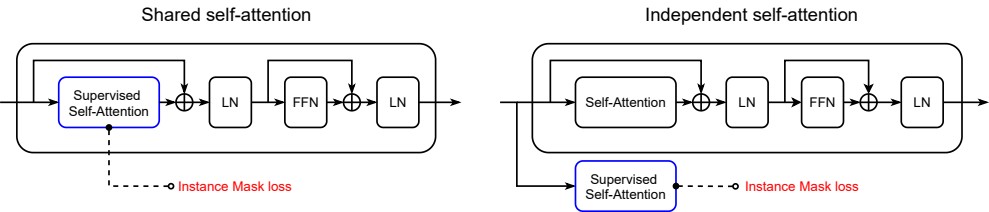

Figure 7: The architecture designs for supervising shared self-attention and independent self-attention.

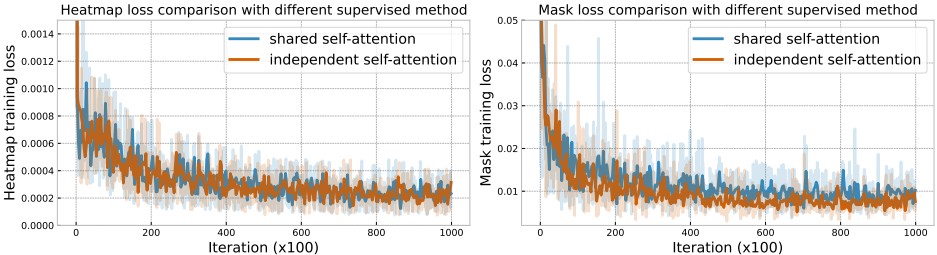

Figure 8: The convergences on the heatmap loss and mask loss when trained with supervising shared self-attention and independent self-attention.

| Supervision type | AP | $AP_{0.5}$ | $AP_{0.75}$ | $AP_M$ | $AP_L$ | AR | $AR_{0.5}$ | $AR_{0.75}$ | $AR_M$ | $AR_L$ |
|---|---|---|---|---|---|---|---|---|---|---|
| Shared | 50.7 | 77.7 | 53.5 | 41.0 | 64.2 | 56.9 | 80.0 | 59.9 | 43.3 | 75.7 |
| Independent | 50.7 | 77.0 | 53.6 | 40.9 | 64.6 | 56.7 | 79.7 | 59.4 | 42.9 | 75.9 |

Table 5: Results on COCO validation set when using shared self-attention and independent self-attention designs.

## A.5 RUNTIME AND COMPLEXITY ANALYSIS

We take the ResNet-101 based model as the exemplar to test two types of grouping algorithm. We use the total 5000 images from COCO validation set. For each image, we run the model forward on a single GPU and the grouping algorithm on the CPU[4], where the grouping runtime is far less than the model forward. In Table 6, we provide a controlled study to compare their differences in the model performance, theoretical complexity for per part assignment, runtime for the whole inference pipeline (keypoint detection, grouping and instance segmentation). Note that the complexity is a theoretical analysis based on the assumption that there are $N$ existing skeletons and $N$ candidates for a certain part type. We report the performances and runtime for the pure bottom-up result and the ones with refinement.

| Grouping Algorithm | Theoretical complexity for per part assignment | AP (BU) | Runtime (BU) | AP (BU+Refine) | Runtime (BU+Refine) |
|---|---|---|---|---|---|
| Part-first view | $\mathcal{O}(N^3)$ | 50.4 | 8.45 img/sec | 65.3 | 5.69 img/sec |
| Body-first view | $\mathcal{O}(N^2)$ | 49.7 | 8.94 img/sec | 64.8 | 5.72 img/sec |

Table 6: Comparison between the body-first and part-first grouping algorithm

In Table 7 we compare our models with the mainstream bottom-up models, in terms of the number of model parameters and computational complexity of the model forward pass. The results of Hourglass (Newell et al., 2017), PersonLab (Papandreou et al., 2018), and HigherHRNet (Cheng et al., 2020) are taken from the HigherNet paper (Cheng et al., 2020). We can see that compared with them, our models have fewer parameters and less computational complexity in the model forward pass.

| Model | Input Resolution | #Param | FLOPs |
|---|---|---|---|
| Hourglass (Newell et al., 2017) | 512 | 277.8M | 206.9G |
| PersonLab (Papandreou et al., 2018) | 1401 | 68.7M | 405.5G |
| HigherHRNet (Cheng et al., 2020) | 640 | 63.8M | 154.3G |
| DEKR (Geng et al., 2021) | 640 | 65.7M | 141.5G |
| Ours (ResNet101+Transformer) | 640 | 45.0M | 102.3G |
| Ours (ResNet152+Transformer) | 640 | 60.6M | 132.7G |
| Ours (ResNet101+Transformer) | 800 | 45.0M | 159.8G |

Table 7: Comparisons on the number of model parameters and model forward complexity.

## A.6 VISUALIZATION FOR HUMAN SKELETONS, INSTANCE MASKS AND KEYPOINT ATTENTION AREAS.

In Figure 9, we visualize the qualitative results predicted by our pure bottom model based on ResNet-152 and $512^2$ input resolution. Our model still can perform relatively well even in some hard cases, such as occluded persons and crowded scene with the existence of a large number of people ($>30$) (shown in the 4-th row in Figure 9). We also can see that the attention areas of the sampled keypoints belonging to a specific person can accurately and reasonably attend to the target person and not attend to the areas excluding the person.

---

[4]NVIDIA Tesla V100 GPU and Intel(R) Xeon(R) Gold 6130 CPU @ 2.10GHz.

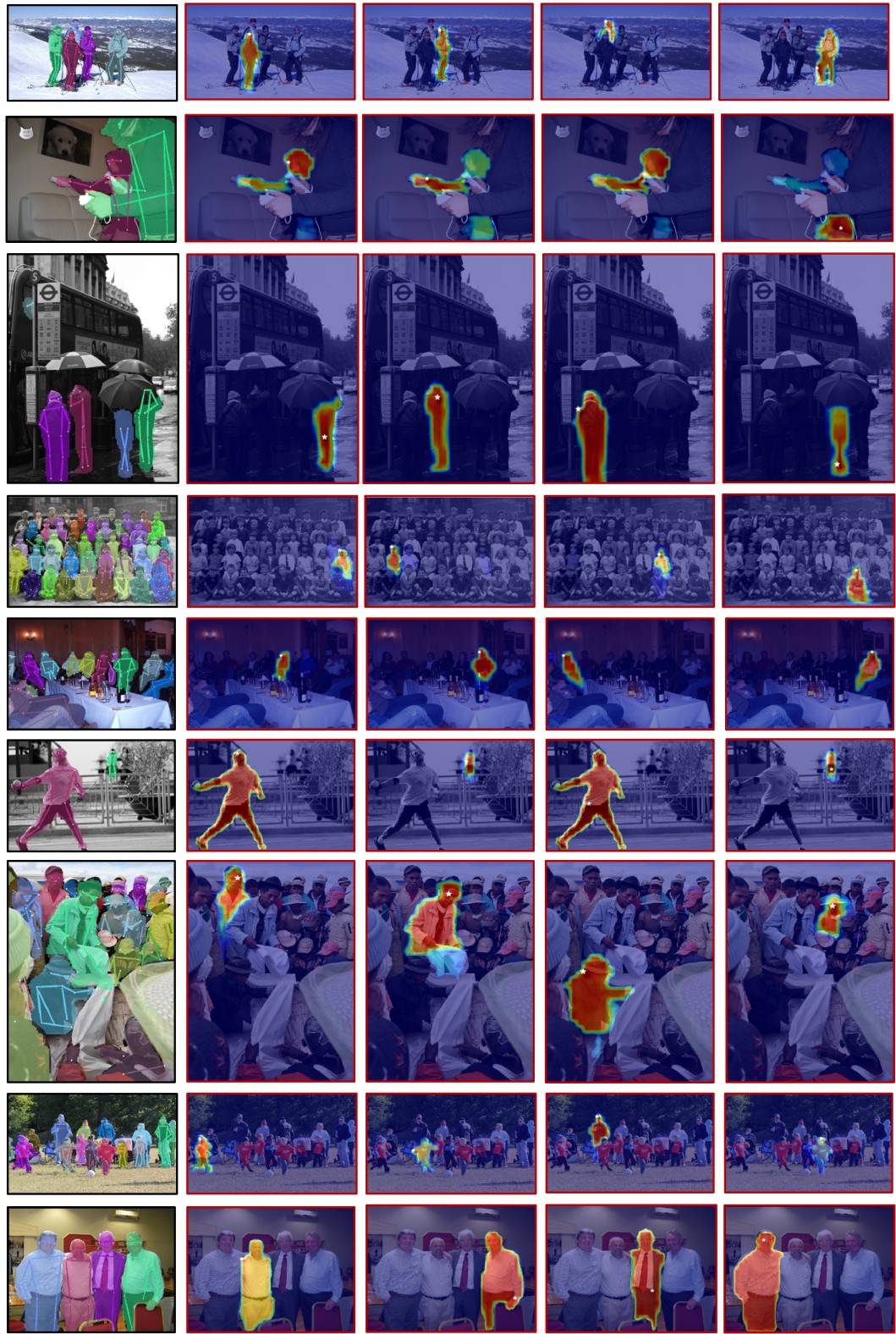

Figure 9: Qualitative visualization results predicted by our pure bottom-up model. For each image, we show the original image plotted with **human poses** and **masks**. And, for each image, we also show **the learned attention areas from the views of 4 sampled keypoints**, each location of which has been annotated by a white color pentagram. Redder areas mean higher attention scores.

