# OpenReview forum: "Attend to Who You Are: Supervising Self-Attention for Keypoint Detection and Instance-Aware Association"
_ICLR.cc/2022/Conference — ICLR 2022 Submitted_

### Official Review · Reviewer_YSvp · 2021-10-27

**Correctness:** 2
**Technical Novelty And Significance:** 3
**Empirical Novelty And Significance:** 2
**Recommendation:** 3
**Confidence:** 4

**Main Review:**

* Strong points.

S1. Clear motivation.
The proposed SSA has a clear motivation. Supervising self-attention maps with human segmentation can guide the self-attention map to focus on correct pixels, which can lead to more accurate pose estimation.

S2. Well-written manuscript.
Overall manuscript and figures successfully deliver their main ideas.

* Weak points.

W1. Lack of numerical comparison between naive self-attention and SSA.
Figure 1 and 6 show qualitative comparison between naive self-attention and the proposed SSA. However, no numerical comparison (i.e., AP on test or validation set) is not reported. This makes me hard to be convinced SSA is clearly more effective than naive self-attention.

W2. Weak experimental results compared to recent state-of-the-art methods.
Table 1 and 2 show comparison between the proposed method and state-of-the-art methods. Compared to other bottom-up methods, the proposed method fails to outperform them. Importantly, most of the bottom-up comparison targets are not recent ones. Recent bottom-up methods, such as [1] and [2], achieve far better results than the proposed one.
[1] achieves AP 66.5 and 68.4 on validation and test-dev set, respectively, without multi-scale testing. [1] and [2] are representative bottom-up methods; however, they are not cited.

[1] Cheng, Bowen, et al. "Higherhrnet: Scale-aware representation learning for bottom-up human pose estimation." CVPR. 2020.
[2] Geng, Zigang, et al. "Bottom-Up Human Pose Estimation Via Disentangled Keypoint Regression." CVPR. 2021.

W3. Unfair comparison using the refinement.
The refinement introduced in Section 3.1 seems used only for the proposed method and not used for other bottom-up methods. The refinement is a kind of post-processing using off-the-shelf well-performing 2D human pose estimator, which is not a novel point of this paper. Applying the refinement only to the proposed method and comparing with other is unfair.

W4. Lack of qualitative results.
No qualitative 2D human pose results are provided.

W5. Is human segmentation the best choice?
Although the human segmentation provides pixel-level annotations, it does not have human articulation information. As human keypoints are connected in the kinematic chain, I guess supervising the attention map with Gaussian heatmap or one-hot matrix could provide better results than human segmentation map since the heatmap and one-hot matrix has activation for each keypoint in channel dimension.


**Summary Of The Paper:**

This paper presents a bottom-up 2D multi-person pose estimation system. The proposed system takes a single RGB image and predicts 2D Gaussian heatmaps of human keypoints, used to localize human keypoints. The main contribution of this paper is that they introduced a supervised self-attention (SSA), which supervises self-attention maps with human segmentation masks. The self-attention maps are computed in Transformer encoder modules, which represents how much each pixel pays attention to all pixels in feature maps. They showed that without SSA, naive self-attention maps tend to have activations on wrong positions.

**Summary Of The Review:**

Although the paper has a clear motivation and is well-written, lack of experimental demonstration and weak experimental results are weak points.

---

> ### Author Response · Authors · 2021-11-18
> **Author response to Reviewer YSvp**
>
> We thank the reviewer for the valuable feedback and hope that our response can address the raising concerns.
>
> > ***Lack of numerical comparison between naive self-attention and SSA ...... However, no numerical comparison (i.e., AP on test or validation set) is not reported. This makes me hard to be convinced SSA is clearly more effective than naive self-attention.***
>
> In the initial version, we did not report the numerical comparison between naive and supervised self-attention due to several considerations. Please refer to **General Response -3** for more details. We would report such results in the revision.
>
> > ***Weak experimental results compared to recent state-of-the-art methods ...... Recent bottom-up methods, such as [1] and [2], achieve far better results than the proposed one ...... [1] and [2] are representative bottom-up methods; however, they are not cited.***
>
> Thanks for your suggestions. We acknowledge that our method achieves a comparable performance with these classic methods but still has a gap with the current SOTA methods. But we think exploiting self-attention to simultaneously solve keypoint detection and association is promising for the bottom-up pose estimation task. In Table 2 of the revision, we would report the results of the SOTA methods including HigherHRNet (Cheng et al., 2020) and DEKR (Geng et al., 2021).
>
> > ***Unfair comparison using the refinement. The refinement introduced in Section 3.1 seems used only for the proposed method and not used for other bottom-up methods. ... ... Applying the refinement only to the proposed method and comparing with other is unfair.***
>
> The typical bottom-up methods like OpenPose (Can et al., 2017), AE (Newell et al., 2017), and Single-Stage Multiperson Pose Machine (Nie et al., 2019) also use a single pose estimator to refine the predictions but they didn't report what concrete rules they use to update the keypoint estimates. And it is difficult to build a completely uniform condition for all the methods. In this paper we use a strict and concrete threshold (OKS>0.75) to reduce the jitter and missing errors in the process of bottom-up detection and grouping. In addition, the refinement in our model also relies on the initially grouped skeletons and predicted masks.
>
> > ***Lack of qualitative results. No qualitative 2D human pose results are provided.***
>
> We show the qualitative results of the estimated pose and masks in the Figure 4 where each image is plotted with the predicted human skeletons and instance masks.
>
> > ***Is human segmentation the best choice?  ...... As human keypoints are connected in the kinematic chain, I guess supervising the attention map with Gaussian heatmap or one-hot matrix could provide better results than human segmentation map since the heatmap and one-hot matrix has activation for each keypoint in channel dimension.***
>
> The human segmentation mask is a good choice but not the only choice. In this work, we use the instance mask to supervise the mean attention area of a person instance by sampling the keypoints locations. This can leverage two types of losses: 1) push loss: positions with 0 values in a person mask can make the keypoints of a person have pairwise low attention scores with the keypoints of other persons; 2) pull loss: positions with 1 values in a person mask can make the keypoints (and non-keypoint locations) belonging to the same person have mutually pairwise high attention scores. We are not sure we have completely understood your proposal. We think it's also feasible to use the GT keypoint heatmaps of a person to supervise the pairwise attention scores between keypoints, as long as we can leverage two types of losses to all pairs of keypoints, as described above.

---

> > ### Comment · Reviewer_YSvp · 2021-11-19
> > **Discussion**
> >
> > Thanks for the clarifications.
> > However, I think W2 and W3 are still my concerns.
> > Table 1 shows that the refinement and post-processing largely boost the accuracy.
> > Although the proposed method relies on such post-processing, it (actually, a little bit large) is beaten by state-of-the-art methods, such as HigherHRNet (CVPR 2020) and DEKR (CVPR 2021), which do not rely on the post-processing.
> > I agree that the motivation of this paper is interesting, but weak experimental results seem weak points of this paper.

---

### Official Review · Reviewer_cESx · 2021-10-29

**Correctness:** 3
**Technical Novelty And Significance:** 2
**Empirical Novelty And Significance:** 3
**Recommendation:** 3
**Confidence:** 4

**Main Review:**

strengths: as far as I know it is the 1st paper to do mid-layer supervision and use instance mask to supervise the transformers attention.

weakness: although they've done several ablation studies, there are not enough
(1) what if the mean of the keypoint is replaced with the mean of vector inside bbox (this seems a more reasonable alternative)
(2) what if this is a multitask transformer, decoded to both instance-segmentation mask and keypoint estimation. Will the end-supervision better be than intermediate supervision?
(3) the final results are not compelling, even worse than PersonLab (2018)
(4) what is the speed and complexity increment  after adding this intermediate loss / for inference (compared with original transformers)
(5) the paragraph above Fig 5 seems not reasonable to me. This makes the method more like a top-down method instead of bottom-up, making this methods less compelling.
(6) Fig 4 need more description to be  better understood.


**Summary Of The Paper:**

The paper proposes to incorporate the instance segmentation task into the human pose estimation task (serving as attention layer loss for transformers). This supervision is novel and serves as a good alternative for AE/PAF, etc. They also propose two decoding methods (part-based / body-based). (with accuracy/time trade off. )

**Summary Of The Review:**

Please refer to the Main Review of the paper.

---

> ### Author Response · Authors · 2021-11-18
> **Author response to Reviewer cESx**
>
> We thank the reviewer for the valuable comments and suggestions and hope that our response can address the raising concerns.
>
> > ***what if the mean of the keypoint is replaced with the mean of vector inside bbox (this seems a more reasonable alternative)***
>
> We're not sure that we have fully understood this question. The goals of sampling all visible keypoints' attention areas and averaging them are 1) to achieve a more smoothed attention area and 2) to build as many pairs of keypoints as possible to leverage the pairwise attention supervision (high and low scores).
>
> > ***what if this is a multitask transformer, decoded to both instance-segmentation mask and keypoint estimation. Will the end-supervision better be than intermediate supervision?***
>
> This architecture is not exactly like a typical multi-task architecture. Our model focuses more on multi-person keypoint detection and grouping. For this question, please refer to the **General response - 4** for more details.
>
> > ***the final results are not compelling, even worse than PersonLab (2018)***
>
> PersonLab uses a larger model (68.7M) and a very high input resolution (1401), resulting in a very expensive model forward complexity (405.5 GFLOPs). Compared with PersonLab, our model has fewer parameters (ResNet-101 based, 45.0M) and less forward inference complexity (159.8GFLOPs under 800 input resolution). We acknowledge that our method achieves a comparable performance with these classic methods but still has a gap with the current SOTA methods. But our method provides a brand-new solution to the bottom-up detection and association problems.
>
> > ***what is the speed and complexity increment after adding this intermediate loss / for inference (compared with original transformers)***
>
> This intermediate loss only works during training. It will not change the architecture of Transformer so that the inference time and complexity will not be increased.
>
> > ***the paragraph above Fig 5 seems not reasonable to me. This makes the method more like a top-down method instead of bottom-up, making this methods less compelling***
>
> The typical bottom-up pose estimation models such as OpenPose (Cao et al., 2017), AE (Newell et al., 2017), and Single-Stage Multi-Person Machine (Nie et al., 2019) also adopt a single pose estimator to refine the results. We follow them and adopt a strict rule to update the keypoint predictions with small localization jitter errors. In addition, the refinement in our model also highly relies on the initially grouped skeletons and predicted masks.
>
> > ***Fig 4 need more description to be better understood***
>
> Thanks for your suggestions. In the revision, we will add more description in the caption of Fig 4.

---

### Official Review · Reviewer_oDDC · 2021-11-02

**Correctness:** 3
**Technical Novelty And Significance:** 2
**Empirical Novelty And Significance:** 2
**Recommendation:** 5
**Confidence:** 4

**Main Review:**

I like that the proposed approach takes advantage of an existing property of the TransPose architecture which is already doing the work of explicitly associating pixels with each other in each of the transformer layers, adding a little extra supervision is straightforward enough and can always be incorporated as dense transformer architecture design matures. Overall, the paper is written clearly and sufficient information is provided to reimplement the method.

However I have some concerns:

- Comparisons are left out to more recent bottom-up human pose work, notably missing is the HRNet family of work such as HigherHRNet (Cheng et al CVPR 2020) and the more recent DEKR (Geng et al CVPR 2021) both of which set a higher bar for bottom-up results and also do not rely on single-person pose refinement. Beating the state-of-the-art is certainly not a requirement here, but it is important to include other recent methods in comparisons.

- It seems to me that the transformer layers introduce serious compute overhead (this is alluded to in the concluding discussion), the paper does not offer much information in terms of the cost of this approach compared to other work. It would be helpful to get some comparisons (e.g. images/sec on fixed hardware, memory requirements, etc).

- A related point is that attention across pixel space requires that a large stride is used to operate at a reasonably low resolution. This is a difficult trade-off in bottom-up human pose where metrics are so sensitive to small localization jitter. It is unclear what good strategies to overcome this might be with this style of architecture.

- Another caveat is the requirement for additional instance segmentation mask supervision which other methods do not rely on. Were any alternatives considered so that this method could be used on datasets where these annotations are unavailable?

- One question I had is how restricting the attention in this way affects accuracy of the predicted heatmaps. From Fig 6 it seems as though it hurts performance somewhat. It would be interesting to know the effect here (for example, by comparing with and without the attention loss and evaluating using oracle assignment)


**Summary Of The Paper:**

This work proposes explicit supervision of transformer attention for the purposes of bottom-up multi-person pose estimation. The network (TransPose [Yang et al. 2021]) passes an image through a resnet followed by several tranformer layers to produce per-pixel heatmaps localizing the joints of all people in the given image. The attention of the transformer layers is across pixel space, so the authors propose supervising the resulting distribution so that the attention score is higher between keypoints belonging to the same person. This is done by sampling the attention masks for keypoints and supervising them with MSE to match the annotated segmentation mask of that person. At test time, detected keypoints can be grouped into invidual people with a simple heuristic parsing of relative attention scores.

**Summary Of The Review:**

The authors propose an nice, simple strategy to associate keypoints in multi-person pose estimation, but I don't know that a compelling enough case is made that this method offers advantages given the restrictions in terms of additional annotation requirements + compute overhead while falling short of and excluding more recent methods in performance comparisons.

---

> ### Author Response · Authors · 2021-11-18
> **Author response to Reviewer oDDC**
>
> We thank the reviewer for the positive feedback and constructive suggestions and hope that our response can address the concerns.
>
> > ***... ... notably missing is the HRNet family of work such as HigherHRNet (Cheng et al CVPR 2020) and the more recent DEKR (Geng et al CVPR 2021) ...... Beating the state-of-the-art is certainly not a requirement here, but it is important to include other recent methods in comparisons.***
>
> We thank the reviewer's reminder. We have added the more recent results of HigherHRNet (Cheng et al., CVPR 2020) and DEKR (Geng et al., CVPR 2021) to the Tab. 2 of the revised paper.
>
> > ***It seems to me that the transformer layers introduce serious compute overhead (this is alluded to in the concluding discussion), ....... It would be helpful to get some comparisons (e.g. images/sec on fixed hardware, memory requirements, etc).***
>
> Thanks very much for your suggestions. Here we report the model parameter number and MACs of our model by using the `thop` library [1] that also supports computing the flops of Transformer. The results are:
>
> | Model                         | Input resolution | #Params | FLOPs  |
> | ----------------------------- | ---------------- | ------- | ------ |
> | Hourglass                     | 512              | 277.8M  | 206.9G |
> | PersonLab                     | 1401             | 68.7M   | 405.5G |
> | HigherHRNet (HRNet-W48)       | 640              | 63.8M   | 154.3G |
> | Ours (ResNet-101+Transformer) | 640              | 45.0M   | 102.3G |
> | Ours (ResNet-152+Transformer) | 640              | 60.6M   | 132.7G |
> | Ours (ResNet-101+Transformer) | 800              | 45.0M   | 159.8G |
>
> The reported results of AE, PersonLab, and HigherHRNet are taken from the HigherHRNet github link [2]. We can see that compared with the typical bottom-up models, our models have fewer parameters and less computational complexity in the model foward process.
>
> > ***A related point is that attention across pixel space requires that a large stride is used to operate at a reasonably low resolution. This is a difficult trade-off in bottom-up human pose where metrics are so sensitive to small localization jitter. It is unclear what good strategies to overcome this might be with this style of architecture.***
>
> Our model is indeed struggling to balance the trade-off between reducing the stride and keeping a large model capacity. Small stride is helpful to reduce the localization jitter and large Transformer model capacity is important to learning good representation for heatmap localization and mask learning. In the current approach, we choose 16 stride for a relatively better trade-off. If choosing a smaller stride like 8, the GPU memory consumption will increase substantially, requiring reducing the input resolution and model size of the Transformer.
>
> > ***Another caveat is the requirement for additional instance segmentation mask supervision which other methods do not rely on. Were any alternatives considered so that this method could be used on datasets where these annotations are unavailable?***
>
> There are alternatives to the use of pixel-level instance mask annotations. When these annotations are unavailable, we also can leverage a loss function only between keypoint locations that encourages the keypoints belonging to the same person to have high attention scores and the ones belonging to different persons to have low attention scores. Please refer to **General Response - 1** for more information.
>
> > ***One question I had is how restricting the attention in this way affects accuracy of the predicted heatmaps ...... It would be interesting to know the effect here (for example, by comparing with and without the attention loss and evaluating using oracle assignment)***
>
> When we used the totally same conditions of the supervised self-attention model, we only achieved 29AP on COCO validation set, which is far from the 50.7AP result achieved by supervising self-attention. This comparison shows the effectiveness and superiority of supervising self-attention for keypoint grouping. For more details, please refer to **General Response -3 and -4**. By the way, what is the method of the "evaluating using oracle assignment"?
>
> [1]: https://github.com/Lyken17/pytorch-OpCounter
>
> [2]: https://github.com/HRNet/HigherHRNet-Human-Pose-Estimation#results-on-coco-test-dev2017-without-multi-scale-test

---

> > ### Comment · Reviewer_oDDC · 2021-11-29
> > **response**
> >
> > I appreciate all the work the authors have put into their responses to the reviewer's comments. While I do not agree with all of the reviewer comments, there is consensus that this paper is not yet ready for acceptance.
> >
> > - There are interesting ideas touched on in this work but I don't think a clear case has been made that intermediate supervision of transformer attention is advantageous compared to prior methods that supervise keypoint associations in the model output.
> > - There is a cost imposed by the spatial attention that restricts further scaling in model size and resolution. While the authors provided useful information in terms of parameter count and FLOPs in their rebuttal, GPU memory is one of the main pain points and this is not reflected in these metrics. In addition, runtime does not necessarily correlate with FLOPs, so we might not have a clear impression of which models are faster during inference/training.
> > - regarding "oracle assignment", the quality of the keypoints can be evaluated by assuming the _best_ possible grouping has been performed, this is made possible by cheating with the ground truth (hence "oracle") the point here is to takeaway the confounding impact of the imperfect association process from the metric so that the keypoints can be judged independently - this can show if the proposed supervision hurts the quality of predicted keypoints or not

---

### Official Review · Reviewer_PtRJ · 2021-11-03

**Correctness:** 3
**Technical Novelty And Significance:** 3
**Empirical Novelty And Significance:** 2
**Recommendation:** 5
**Confidence:** 3

**Details Of Ethics Concerns:**

None.

**Main Review:**

**Strengths**

The idea to incorporate self-attention models to extract semantic features for keypoints association is a novel and elegant approach to address this problem. The efficacy of this approach is also verified by the experimental results. For large objects, the neighboring keypoints could be several hundred pixels away and Personlab [1] adopt recurrent offset refinement to refine the inaccurate regressed long-range offsets. In contrast, the proposed model benefit from the self-attention and can easily group keypoints with large offsets.

Another contribution of this work is to unify keypoint detection, grouping and segmentation. The instance mask helps to supervise the association between keypoints of a single instance and the predicted keypoint detection would help produce better masks from a bottom-up perspective. We can see the two tasks are highly-correlated and the proposed approach seems to be a reasonable and novel design in this direction.

**Concerns**

Supervising self-attention with instance mask is a natural idea to assist keypoint grouping but there are some concerns related to the choice of supervision that are not addressed well. The visualization of the association reference in Figure 1 show that the supervised self-attention helps to discriminate keypoints from the same instance and from different instances. However, we could also see that the naive self-attention produce self-attention scores with good characteristics that start by looking at the context around the instance and then shift to focus on keypoints of the same instance. Meanwhile, the supervised self-attention seems to only focus on associating keypoints of the same instance, which is a direct result of the mean squared error (MSE) of the instance mask loss. It seems that the self-attention behavior is completely changed by the instance mask loss and subsequently the final keypoint heatmap. It is good to produce a instance segmentation mask at certain layer but it is unclear how will this affect the keypoint features and the keypoint detection. Will the instance mask loss make the keypoint features less discriminative and hence less accurate keypoint localization? There is no ablation study experiments on the design of the instance mask loss to address such concerns so the advantage of such loss is doubtful beyond the visualization results. Ablation study in A.3 show that there is no clear differences between supervising different layers of the self-attention model, which I don't understand since the self-attention scores focus on very different regions before and after the layer that is supervised.

Experimental results are limited and fail to show the benefits of self-attention over regression. Most quantitative results are only roughly comparable with previous methods even with different refinements. The issue of Transformer models performing worse on smaller objects (e.g., DETR [2]) is also left unaddressed. While the proposed model can elegantly unify the task of keypoint detection and instance segmentation, the advantage of such approach is not supported by the experiments. The experiments cannot show the benefits of sharing the features for both keypoint detection and instance segmentation and there are no known superior aspects of the proposed approach in terms of number of parameters, MACs, or post-processing time.

**Summary Of The Paper:**

Bottom-up methods tackle the problem of multi-person detection, pose estimation, and segmentation by localizing human keypoints and then grouping them into person instances, which inevitably bring up the question of what features to use for grouping and how to efficiently group keypoints and break down corner cases. In this work, the authors propose to use Transformer to exploit associative information between keypoints. Initial experiments show that naive self-attention has non-zero attention scores between keypoints from different human instances, and even high attention scores in crowded scenarios. Such self-attention layers make it hard to directly associate keypoints of the same instance from the self-attention scores, and subsequently requires careful hyperparameter tuning and inevitably introduce errors in hard examples. Therefore, the authors propose a instance mask loss to supervise self-attention with instance segmentation masks.

Another advantage of such framework is to unify the task of keypoint detection and instance segmentation. The segmentation mask can be easily generated from the self-attention scores and the network learns a representation that is consistent with both the keypoint locations and the instance masks.

**Summary Of The Review:**

Overall, I think this paper presents a novel and interesting idea. However, the current experimental results are relatively weak and cannot show the strength of the method, which limits the contribution. There are also some concerns with the self-attention supervision that are not explained well and not supported by the ablation study.

---

> ### Author Response · Authors · 2021-11-18
> **Author response to Reviewer PtRJ.**
>
> We thank the reviewer for the detailed feedback and constructive suggestions and hope that our response can address the concerns.
>
> > ***However, we could also see that the naive self-attention produce self-attention scores with good characteristics that start by looking at the context around the instance and then shift to focus on keypoints of the same instance. Meanwhile, the supervised self-attention seems to only focus on associating keypoints of the same instance, ... ... It seems that the self-attention behavior is completely changed by the instance mask loss and subsequently the final keypoint heatmap.***
>
> We appreciate your detailed observations. The naive self-attention layers indeed gradually focus on the keypoints of the same instance, while they also generate false attention responses on the keypoints belonging to other person instances (Fig. 1). We conjecture that this phenomenon is because the model will respond to the keypoints with similar features, due to that is has not learned discriminative features between different person instances.
>
> You're right. The self-attention behavior is completely changed by the instance mask loss, but this change is just what we need and expect and this is beneficial to the keypoint grouping. It's also biased to directly say that this change does harm to keypoint localization, because we have adopted many tricks to overcome this issue ，and the situation of loss convergences (Fig. 6) where the small sacrifice in heatmap loss fitting brings big benefit in mask loss fitting also confirm this point.
>
> > ***It is good to produce a instance segmentation mask at certain layer but it is unclear how will this affect the keypoint features and the keypoint detection. Will the instance mask loss make the keypoint features less discriminative and hence less accurate keypoint localization? ....***
>
> We appreciate your insightful comments. Figuring out how intermediate instance mask loss will affect keypoint features and keypoint detection is also interesting and important for our method. We first visualize the qualitative differences between naive self-attention and supervised self-attention (such as in Fig.1). And then in section 3.2, we make an ablation on the quantitative differences by comparing their convergences. As suggested by the reviewer, we should report numerical comparisons in performance to more accurately verify the effectiveness of our method. Please refer to **General Response - 3 and -4** for more details.
>
> > ***Ablation study in A.3 show that there is no clear differences between supervising different layers of the self-attention model, which I don't understand since the self-attention scores focus on very different regions before and after the layer that is supervised.***
>
> In Fig. 1, we show attention patterns in all attention layers, where the 4-th attention layer is supervised. If we choose other layers to supervise, the distribution of attention in different layers will be in another way.
> In addition, we have to admit that, the attention pattern changes in different transformer layers are still beyond our understanding to some extent, since most of the self-attention matrices in different layers are mainly computed in an unsupervised manner and there are still residual connections parallel to the qkv self-attention computation that adaptively change the distribution of attention. However, the self-attention matrix supervised by our target will be what we expect.
>
> > ***Most quantitative results are only roughly comparable with previous methods even with different refinements. The issue of Transformer models performing worse on smaller objects (e.g., DETR [2]) is also left unaddressed.***
>
> We mainly compare our method with OpenPose, AE, and PersonLab. OpenPose and AE also use a single pose estimator to refine the predictions and they didn't report what concrete rules they use to update the keypoint estimates. So it is hard to set a uniform standard for all the methods. In this paper we use a strict and concrete threshold (OKS>0.75) to refine the keypoints (reducing the jitter and missing error of bottom-up detection and grouping). And the refinement also relies on the initially grouped skeletons and predicted masks. Yes, the poor performance on small objects is an issue for the Transformer model due to the limited input sequence length. We expect future works can address this problem.
>
> > ***While the proposed model can elegantly unify the task of keypoint detection and instance segmentation, ... ...  in terms of number of parameters, MACs, or post-processing time.***
>
> Thanks for your approval. We think our method gives a brand-new solution to unify keypoint detection, grouping, and instance segmentation, and there is still a lot of potential to optimize and improve performance. in Appendix 5 of the revised version, we report the overall inference time of the whole pipeline and we will add the MACs and parameters comparison.

---

> > ### Comment · Reviewer_PtRJ · 2021-11-29
> > **Re: Author response to Reviewer PtRJ**
> >
> > Thanks for the clarification and the additional experimental results. The quantitative results are interesting and resolves some of my concerns regarding the joint prediction of keypoints and instance masks.
> >
> > Overall I think this work proposed an interesting idea to learn self-attention maps and associate keypoints of the same instance, which is important for bottom-up methods. The proposed approach seems intuitive and somewhat novel, but the current quantitative results are not promising enough. Using self-attention models for keypoint association seems reasonable but certain designs of the framework may be questionable and limit the overall performance, which prevent the reviewer from scoring it higher in its current state.

---

### Official Review · Reviewer_mi8h · 2021-11-03

**Correctness:** 4
**Technical Novelty And Significance:** 2
**Empirical Novelty And Significance:** 2
**Recommendation:** 3
**Confidence:** 4

**Main Review:**

* Strengths:
    * The idea is clear and reasonable. Self-attention module computes attention for every point in the feature map. It is reasonable to restrict the region where a self-attention module puts its attention by using some prior knowledge. The authors proposed two methods to restrict the attention region, one is a loss applied on attention map  which is interesting, and the other is attention-score based keypoint grouping. The visualized attention maps show that the proposed loss works well.
    * The ablation study on Tab.1 shows the effectiveness of the proposed method.
* Weaknesses:
    * The idea is lack of novelty. The restrict attention region is reasonable but not novel, since the core idea is still instance segmentation.
    * The performance is not promising when compared with some classic methods.
    * As for instance segmentation, the authors argue that the worse performance is due to lower resolution. I think it is necessary to set up a fair experiment, otherwise the comparison is not very meaningful.
    * In 3.2, only loss is shown, but it is not clear how the supervised self-attention contributes to accuracy.


**Summary Of The Paper:**

This paper proposes a bottom-up multi-person pose estimation method based on transformer model. In this method, a new loss is used to encourage a self-attention map to put its attention on the region of an instance via the given ground-truth segmentation mask. Besides, the attention scores between estimated keypoint-locations are used to assign the estimation to correct instance.

Experiments are applied on COCO keypoint dataset. The results shows that the proposed method outperforms or is comparable with some competitors such as OpenPose and Person Lab on keypoint detection. When compared with the existing methods which explicitly estimate instance segmentation mask, the proposed method performs worse.

**Summary Of The Review:**

The idea is clear and reasonable, but lack of novelty and promising performance.

---

> ### Author Response · Authors · 2021-11-18
> **Author response to Reviewer mi8h**
>
> We thank the reviewer for the valuable comments and hope that our response can address your concerns.
>
> > ***The idea is lack of novelty. The restrict attention region is reasonable but not novel, since the core idea is still instance segmentation***.
>
> We'd like to clarify that the instance segmentation task is not our main goal and the novelty of this method is to exploit the 0/1 values of instance mask to ensure the detected keypoints can have pairwise high or low attention scores, which is recognized by other reviewers as novel and elegant. To our best knowledge, our method is the first one to supervise the intermediate transformer layer for keypoint detection and grouping. As for use of the instance mask, please refer to **General Response -1 and -2** for more details.
>
> > ***The performance is not promising when compared with some classic methods***.
>
> To address the multi-person pose estimation in a bottom-up way, the classic methods like OpenPose, Associative Embedding, and PersonLab are highly optimized and often set as baselines for the follow-up works. We acknowledge that our method achieves a comparable performance with these classic methods but still has a gap with the current SOTA methods. But our method provides a brand-new solution to the bottom-up detection and association problems, the idea of which is promising for multi-instance pose estimation and can be applied to other vision tasks like object detection and instance segmentation.
>
> > ***As for instance segmentation, the authors argue that the worse performance is due to lower resolution. I think it is necessary to set up a fair experiment, otherwise the comparison is not very meaningful.***
>
> PersonLab only reported the instance segmentation performance under $1401^2$ input resolution and 8 strided ResNet backbone. Such a choice is feasible for the fully convolutional architecture but impracticable for Transformer architecture because the input sequence length will increase to be very long which is prohibitively expensive for training and inference. In the current approach, we choose 16 stride for a relatively better trade-off. If choosing a smaller stride like 8, the GPU memory consumption will increase substantially, inevitably requiring us to reduce the input resolution and model size of the Transformer. So it is still hard to build a completely fair condition to compare our transformer-based model with the FCN-based model. We might add a specific mask branch (as described in General Reponse -2) to refine the mask prediction in future works.
>
> > ***In 3.2, only loss is shown, but it is not clear how the supervised self-attention contributes to accuracy***
>
> Thanks for your advice. As for the numerical performance comparison, please refer to **General Response - 3** for more information.

---

### Author Response · Authors · 2021-11-15
**General response: Reply To all the reviewers for the common concerns**

We thank all the reviewers for their efforts in reading and reviewing. These comments and suggestions have significantly helped us to improve the paper. Considering there are similar concerns from the reviewers, we summarize them into several points for better clarifying, and then we will reply to each reviewer point-to-point.

### 1.The use of pixel-level instance mask annotations

We’d like to point out that, the initial/core spirit of the proposed method is to use some type of constraint terms to control the behaviors the self-attention, which is independent of the necessity of the instance segmentation or the use of the pixel-level annotations. Such types of constraint terms can be supervision or self-supervision signals, as long as they can ensure that the detected keypoints belonging to the same or different instances could have pairwise higher or lower attentions.

In this work, why we choose the instance mask as the supervision signal lies in that the ***0-value/1-value*** positions in a GT person instance mask can directly supervise the keypoints of a person to have ***lower/higher attentions*** with the areas ***excluding/including this person’s body area***. The values in instance masks provide an ideal distribution for the instance association so that using instance masks is straightforward and effective.

We acknowledge that using the pixel-level mask annotations is a little bit redundant, and such annotations may not be available for some other datasets. But even under such conditions, our idea is still applicable as we can enforce the constraint loss only between keypoint locations, i.e., a push&pull loss like AE (Newell et al., 2017) in a location-to-location way instead of a location-to-region way, thus avoiding the use of instance mask.

### 2. The performance on the instance segmentation

Improving the performance of the instance segmentation task is not our main goal. In this work we just use the coarse interpolated attention map as the final instance segmentation results, rather than using any extra neural network or mask branch to refine the mask predictions, which is often considered as a crucial step to improve the instance segmentation accuracy, such as in Mask-RCNN (using a mask head with many convolutional layers) or in PersonLab (using a background/foreground segmentation head and a long-range offset prediction head). Our design for instance segmentation is still in a naïve mode and we believe further optimization could improve the performance of instance segmentation.

### 3. Quantitative Performance comparison between naïve self-attention and supervised self-attention

Although we experimented with using naïve self-attention patterns to group keypoints and tested the model performance under a fairly controlled condition, we did not report the performance on COCO and make the numerical comparisons in the initial version due to the following considerations:
1.	**Non-deterministic grouping strategy and ineffectiveness for many cases**. If we use the naïve self-attention scores, it’s difficult to find a good strategy to determine which attention layers should be chosen as the grouping reference. Some layers may be suitable for grouping but some may bring huge noise and interference, which makes keypoint grouping highly dependent on specific experimental observations. To make matters worse, for many hard cases, each of the attention layers cannot effectively distinguish different person instances based only on the attention scores. As a result, false-positive attention highlight will cause wrong connections, inevitably destroying the model performance.
2.	**Numerical performance on COCO**. Actually, we also experimented with using the naive attention matrix to group keypoints, by averaging the attentions from all transformer layers. When we used the totally same conditions (including model configuration, training & testing settings and grouping algorithm) of the supervised self-attention model - (res152, s16, i640), we only achieved ***29AP*** on COCO validation set, which is far from the ***50.7AP*** result achieved by supervising self-attention. This comparison shows the effectiveness and superiority of supervising self-attention for keypoint grouping. We thank the reviewers for the suggestions on reporting the numerical results since these results can answer the effectiveness of the proposed method. We would report it in the revision.

---

> ### Author Response · Authors · 2021-11-15
> **General response (continued): Reply To all the reviewers for the common concerns**
>
> ### 4. Will the intermediate instance mask loss affect the keypoint features and the accuracy of keypoint localization? Will an independent branch with end-supervision be better than intermediate supervision?
>
> We appreciate the reviewers for their questions and suggestions on the impacts brought by introducing the instance mask loss as the intermediate supervision.
>
> First of all, we have to say that adding some subjective assumptions and constraints to the self-attention matrix that was originally computed in an unsupervised manner will inevitably affect the keypoint localization to some extent. But it is not easy to directly draw a conclusion about whether this impact is negative or positive for keypoint localization. In this paper, considering the COCO training data is abundant and overfitting is not an issue, we think that the fitting error on training data can indirectly reflect the quality of model learning on heatmap localization and mask prediction. Thus, in Sec 3.2 and Fig. 6, we compare the convergences of heatmap and mask loss. Fig.6 shows that supervising the intermediate self-attention does bring a little negative effect to the heatmap learning (may lead to keypoint features being less discriminative). But this kind of sacrifice is acceptable and cannot be avoided, because without imposing such constraints, using naïve attention cannot achieve instance-discriminative, and the attention-based grouping will not work (29AP on COCO) in this situation. In another word, we achieved big gains with small sacrifices.
>
> In this work we also adopt several strategies to minimize the impact of the instance mask loss on the keypoint heatmap localization as much as possible. (1) We experimented with different weight ratios to find a better one to balance the heatmap loss and mask loss. We finally set the weight ratio to 1:0.001 for the heatmap loss and mask loss for a better trade-off; (2) In Appendix 3, we study the effects of supervising self-attention at different layer depths. Although the differences are not very obvious, the results show supervising the penultimate or third-to-last layer attains a better performance. These phenomena can also be explained. Adding instance mask loss at first several transformer layers will make subsequent layers less discriminative to keypoint features. *The existence of the residual path parallel to the supervised self-attention layer may also adaptively reduce the effect of the instance mask loss on the subsequent transformer layers*, since we only leverage the sparse constraints to the self-attention matrix in a certain transformer layer. Besides, there are the transposed convolutional layers appended to transformer (shown in Fig. 2) which is helpful for keypoint heatmap learning.
>
> In fact, we also conducted an ablation on whether using an extra query-key attention head inserted in the intermediate transformer layer will be helpful to the keypoint localization and mask learning. Such an independent query-key attention head can be seen as a parallel branch to leverage an “end-supervision” with instance mask loss. However, we find that such a design achieves a very similar performance on COCO validation set, compared with using a shared qkv self-attention layer for both tasks. The results are:
>
> | Method    | AP                     | Ap .5 | AP .75 | AP (M) | AP (L) | AR    | AR .5 | AR .75 | AR (M) | AR (L) |
> | :----------------------------------------------------------- | ------------------------------------------------------------ | -----| ------ | ------ | ------ | ----- | ----- | ------ | ------ | ------ |
> | shared self-attention|50.7 |77.7| 53.5| 41.0| 64.2| 56.9| 80.0| 59.9| 43.3| 75.7|
> |Independent self-attention|50.7 | 77.0 | 53.6 | 40.9 | 64.6 | 56.7 | 79.7 | 59.4 | 42.9 | 75.9|
>
> In addition, the fitting on the heatmap loss shows little difference under such two different designs. Such results further validate that introducing an intermediate instance mask loss generates a weak effect on the prediction of keypoint heatmaps. We thank the reviewers for the suggestions on such comparisons and we will add this ablation study in the appendix (Appendix 4) of the revised version.

---

### Decision · Program_Chairs · 2022-01-20

**Decision:**

Reject

**Comment:**

This paper proposes a bottom-up multi-person pose estimation method using a Transformer model. There is consensus among the reviewers that this paper is not ready for acceptance/publication. Although some reviewers find the proposed idea interesting (some find it lacking novelty though), all the reviewers agree that the quantitative experimental results are not promising. Some reviewers explicitly criticized lacking empirical accuracy compared to state-of-the-arts. The authors provided additional details and results in the rebuttal, but they were not sufficient to change the opinions of the reviewers.

We recommend rejecting the paper.